# Time-to-Death approach in revealing Chronicity and Severity of COVID-19 across the World

Vivek Verma[1]*, Ramesh K. Vishwakarma[2], Anita Verma[1], Dilip C. Nath[3], Hafiz T. A. Khan[4]

1 Departments of Neurology, All India Institute of Medical Sciences (AIIMS), New Delhi, India, 2 Department of Biostatistics and Bioinformatics, King Abdullah International Medical Research Center/King Saud bin Abdulaziz University for Health Sciences-MNGHA, Riyadh, Saudi Arabia, 3 Assam University, Silchar, Assam, India, 4 The Graduate School, University of West London, Ealing, London, England, United Kingdom

* viv_verma456@yahoo.com

**Data Availability Statement:** The data underlying the results presented in the study is available at Our World in Data (https://ourworldindata.org/coronavirus-source-data).

## Abstract

### Background

The outbreak of coronavirus disease, 2019 (COVID-19), which started from Wuhan, China, in late 2019, have spread worldwide. A total of 5,91,971 cases and 2,70,90 deaths were registered till 28th March, 2020. We aimed to predict the impact of duration of exposure to COVID-19 on the mortality rates increment.

### Methods

In the present study, data on COVID-19 infected top seven countries viz., Germany, China, France, United Kingdom, Iran, Italy and Spain, and World as a whole, were used for modeling. The analytical procedure of generalized linear model followed by Gompertz link function was used to predict the impact lethal duration of exposure on the mortality rates.

### Findings

Of the selected countries and World as whole, the projection based on 21st March, 2020 cases, suggest that a total (95% CI) of 76 (65–151) days of exposure in Germany, mortality rate will increase by 5 times to 1%. In countries like France and United Kingdom, our projection suggests that additional exposure of 48 days and 7 days, respectively, will raise the mortality rates to10%. Regarding Iran, Italy and Spain, mortality rate will rise to 10% with an additional 3–10 days of exposure. World's mortality rates will continue increase by 1% in every three weeks. The predicted interval of lethal duration corresponding to each country has found to be consistent with the mortality rates observed on 28th March, 2020.

### Conclusion

The prediction of lethal duration was found to have apparently effective in predicting mortality, and shows concordance with prevailing rates. In absence of any vaccine against COVID-19 infection, the present study adds information about the quantum of the severity

**Funding:** The author(s) received no specific funding for this work.

**Competing interests:** The authors have declared that no competing interests exist.

and time elapsed to death will help the Government to take necessary and appropriate steps to control this pandemic.

## 1. Introduction

There are mainly three epidemics, which arisen with infections related to emerging coronaviruses in last two decades, which are namely Severe Acute Respiratory Syndrome (SARS)and the Middle East Respiratory Syndrome (MERS) and coronavirus disease-19 (COVID-19). It is a respiratory syndrome related with the coronavirus that was first identified in Hubei Province, PRC, in late 2019 and have spread globally, with the World Health Organization (WHO) declared it a pandemic on 11 March 2020. Till 21st, March, 2020, this outbreak has affected approximately300,000 people across the globe. Coronavirus is basically a zoonotic virus [1], which is commonly found in camel, rat, bat and cattle. In previously known strains of Coronavirus, viz., MERS-Cov and SARS-Cov [1, 2], it was rarely found that an animal coronavirus infected human beings and through which it spreads to community. But, the infectious nature of the virus to human being was firstly observed when its new strain, COVID-19infected people were admitted to hospitals with similar problems of pneumonia with an unknown etiology [3]. The chronicity and fatality of COVID-19 can be visualized from the fact that it is spreading among humans and affects respiratory system, which causes severe respiratory diseases and leads to death [4–6]. The threat due to COVID-19 can be seen from the fact that due to increase and spread of its related cases and deaths, WHO on 11th of February, 2020 declared the current time as a global public health emergency situation [7]. According to Centre for Disease Control and Prevention (CDC), COVID-19 is also acting as a betacoronavirus like its earlier strains of MERS-Cov and SARS-Cov, which were originated from bats [4].

The spreading kinetics of COVID-19 can be framed through cases found in China. Initially, during 18-29th December, 2019, due to COVID-19 there were only 5 confirmed cases reported and among them only single death occurred. The infected cases on 25th January 2020 were increased to 1975,out of which 56 were deceased, and the count raised to 7734 with 90 deaths till 30thJanuary, 2020 and spread to globally with fatality rate of 2.2% approximately [8]. According to WHO report, Italy is the worst hit with 9136 deaths, following Spain 4858 and China a 3301 on 28th, March, 2020. Other mostly affected list of countries includes Germany, Iran, France and the United Kingdom- where Prime Minister Mr. Boris Johnson has been tested positive.

The most common pathological symptom observed among COVID-19 [4] infected individual is that the virus damages the alveolar, which leads to a respiratory failure and as normal like flu, fever, cold, cough and shortness of breath, along with them the other severe symptoms observed are sputum production, haemoptysis, lymphophenia and pneumonia, in some cases increasing dyspnea and hypoxemia in the upper lobe of the lung were also observed. The silent feature of COVID-19 is its associated symptoms that they will appear during incubation period of 2–14 days [4]. COVID-19 can infect individuals of all ages and genders and can spread easily from one person to another, but the likelihood of getting infected is higher among older population, on various medical conditions, such as, diabetes, cardiovascular diseases, hypertension, cancer and chronic respiratory diseases [9, 10]. Severe illness due to the COVID-19 leads to death (mortality rate of 3% approximately) [10].

In the genomic sequence analysis, the COVID-19 has found to similar (82% of SARS-Cov and 50% of MERS-Cov) [4] to earlier coronaviruses, which indicates mammals are more likely

link between COVID-19 and humans [4]. The mode of transmission of COVID-19 from the infected individual to another person or community is through individual's droplets spread while coughing and sneezing [4]. As of now, no vaccine or anti-viral drugs is available against COVID-19 infection for potential therapy. Therefore, precautions, self-sanitization and isolation, and practicing social distancing are only alternatives measures that are recommended to control the current outbreak. In order to minimize the transmission and spreading of COVID-19, 'community-wide containment' [11] is also required to be implemented on proper time.

Majority of the recent investigations discussed in recent literatures were emphasized to provide a basis to understand the etiological, genetically and geographical aspects in association with COVID-19 virus. As this is an ongoing issue and there is no research available on the measurement of the severity of COVID-19 and time elapsed to death. Intensity of death may vary due to demographic, socioeconomic, life-style factors and social norms. It is thus important to know the risk factors of propensity of deaths due to deadly COVID-19. Based on the available information, long period of exposure and systematic increase in number of cases are also increasing its associated deaths. Thus the study aims to examine it here to add knowledge in the exiting literature.

To study the COVID-19 virus spreading and fatal kinetics, it is assumed that individuals of each country have their own tolerance, beyond which they start dying due to lethal characteristics of the virus. The total number of deaths occurred due to COVID-19 after a particular duration of exposure may considered as a binomial random variable with known number of its infected cases, but having unknown probability of death occurrence on that day, say $\pi$. If $\pi$ can be estimated by using the available information about the infection due to COVID-19 virus, then one can use this estimate to derive the duration of infection that is estimated to cause $\pi$ proportion of death among infected individuals.

### 1.1 Objective

The objective of the present study is to suggest an effective and efficient model that captures the predicted duration of exposure to COVID-19 and its impact on the mortality rates, $\pi$, increment. And to evaluate the choronicity of COVID-19 in different countries, who are in critical state by using the kinetics of time to death and define lethal duration for death occurrence.

## 2. Methods

### 2.1 Data description

The data was initially accessed on 21th March 2020from the website of Our World in Data (https://ourworldindata.org/coronavirus-source-data), which presented data from reliable sources of WHO and European Centre for Disease Prevention and Control (ECDC), for modeling purpose and again accessed on 28th March 2020 for validation of suggested methodology. This dataset organizes epidemiological aspects of incidence and mortality due to COVID-19, as reported by public health authorities worldwide. For the present work, we have considered the date-wise infected and death cases due to COVID-19of seven countries viz., Italy, Spain, China, Iran, France, United Kingdom and Germany, where highest number of deaths have been reported, and World as a whole. Here, the first date of reporting of COVID-19 case, corresponding to each selected country, was considered as the starting day. For the prediction of lethal duration to attain a particular mortality rate, country's or World's information after the 10th day of reporting was used. Based on mortality rates reported by each country and World on 21st March 2020, countries are classified into three categories, viz., Category

I for countries whose mortality rates are less than 2%, Category II for countries whose mortality rates lies between 2–5% and Category III for countries where mortality rates reached more than 5%. Based on the mortality rates, Germany belongs to Category I, China, France, United Kingdom fall under Category II, and Iran, Italy and Spain included in category III. As per categories, the predicted mortality rates were also grouped into three classes, Class 1 of mortality rates 1–5% (with 1% increment), Class 2 of 6–10% mortality rates (with 1% increment) and Class 3 of 20–50% mortality rates (with 10% increment).

Let $\pi$ denotes the mortality rates due to infection in the population after specific duration of exposure and $LD_\pi$ symbolizes the 'lethal duration' for severity due to COVID-19 in a population, which is estimated to cause $\pi$ proportion of death among infected individuals. Here, we have tried to estimate and examine chronicity due to lethal duration of exposure, $LD_\pi$, and severity due to COVID-19 virus, by using the available information of total number of cases and deaths till date. The model considered for the present analysis is based on the basic assumptions that (a) mortality rate is a function of duration of exposure to the infection (b) mortality rates across the countries is independent. The methodological aspect of $LD_\pi$ estimation is done using Gompertz link function and its related 95% confidence interval is discussed in Appendix section. The analysis was performed using SAS software, version 9.4 (SAS Institute Inc, Cary, North Carolina).

## 3. Results

Tables 1 and 3 presented the summary of total number of cases and deaths occurred due to COVID-19, and duration of exposure, till 21st and 28th March 2020, respectively, along with that the observed mortality rate in different countries and World as a whole, and classification of these probabilities in different categories. Majority of the COVID-19 infected cases found as of 21st March, 2020 were from China (81416) followed by Italy (47021) then Spain (19980), and in terms of number of deaths occurred, Italy was on top followed by China and Iran. On observing the number of cases and deaths occurred due to COVID-19 as of 28[th] March, 2020 majority were from Italy (86498) followed by China (82213) then Spain (64059), and in the similar order of number deaths reported. Table 2 presented the predicted lethal duration, $\widehat{LD}_\pi$ (in days of exposure) with 95% confidence interval (CI), for death occurrence in different countries due to COVID-19 for the given mortality rate ($\pi$). The previous Class 1 and 2 of Category II and III were showing the pattern of that transition. The visualization of the predicted lethal duration (days) based on the classes of mortality rates (probability), which is expected to be followed by each selected country and world as a whole based on their categories and with the 95% CI were depicted in Figs 1 and 2–4, respectively.

**Table 1. Duration of exposure and mortality rates in different countries due to COVID-19 on 21[st] March, 2020.**

| Country | Total Cases | Total Deaths | Duration of exposure (21[st] March, 2020) | Mortality rate $\pi$ | Category[#] |
|---|---|---|---|---|---|
| Germany | 18323 | 45 | 54 | 0.002 | I |
| China | 81416 | 3261 | 81 | 0.040 | II |
| France | 12612 | 450 | 57 | 0.036 | II |
| United Kingdom | 3983 | 177 | 51 | 0.044 | II |
| **World** | **271364** | **11252** | **81** | **0.041** | **II** |
| Iran | 19644 | 1433 | 31 | 0.073 | III |
| Italy | 47021 | 4032 | 51 | 0.086 | III |
| Spain | 19980 | 1002 | 50 | 0.050 | III |

# Category I if $\pi \leq 2\%$; Category II if $\pi$ (2–5)%; Category III if $\pi > 5\%$;

**Table 2. Predicted lethal duration, $\widehat{LD}_\pi$ $(t)$ (in days), in different countries due to COVID-19 for the given probability of death ($\pi$).**

| Class | Probability ($\pi$) | Predicted lethal duration, $\widehat{LD}_\pi$ exposure, for death occurrence in countries* (in days) with 95% CI | | | | | | | |
|---|---|---|---|---|---|---|---|---|---|
| | | Category I | Category II | | | | Category III | | |
| | | Germany | China | France | UK | World | Iran | Italy | Spain |
| 1 | 0.01 | 76(65–151) | 15(13–17) | 33(9–40) | 39(37–41) | 15(13–16) | 10(9–11) | 18(15–20) | 34(32–35) |
| | 0.02 | 91(42–248) | 33(31–35) | 46(62–50) | 44(43–45) | 34(32–35) | 15(14–16) | 25(23–27) | 40(39–41) |
| | 0.03 | 101(77–332) | 53(51–55) | 57(54–67) | 47(46–48) | 55(54–56) | 19(18–20) | 30(28–32) | 44(43–45) |
| | 0.04 | 109(81–410) | 74(72–76) | 65(60–108) | 50(49–51) | 78(76–80) | 22(21–23) | 34(32–36) | 47(46–48) |
| | 0.05 | 116(84–482) | 96(92–101) | 73(64–158) | 52(51–53) | 102(99–107) | 25(24–26) | 38(37–40) | 50(48–51) |
| 2 | 0.06 | | 119(112–127) | 80(67–216) | 53(52–55) | 128(121–136) | 28(27–29) | 42(41–43) | 52(51–53) |
| | 0.07 | | 143(132–156) | 87(73–284) | 55(53–57) | 155(145–166) | 30(29–31) | 45(44–46) | 54(53–55) |
| | 0.08 | | 167(153–186) | 93(73–358) | 56(54–58) | 183(169–199) | 33(32–34) | 48(46–49) | 56(55–58) |
| | 0.09 | | 193(174–217) | 99(77–441) | 57(55–60) | 211(194–233) | 35(34–37) | 51(50–52) | 58(56–60) |
| | 0.10 | | 218(196–249) | 105(79–532) | 58(56–62) | 241(220–269) | 37(36–39) | 54(52–56) | 59(58–61) |
| 3 | 0.20 | | | | | | 56(53–60) | 76(72–82) | 71(67–75) |
| | 0.30 | | | | | | 73(68–79) | 95(88–105) | 79(74–86) |
| | 0.40 | | | | | | 88(81–97) | 113(102–128) | 87(81–95) |
| | 0.50 | | | | | | 104(95–116) | 130(116–151) | 93(86–103) |
| | P value | 0.0021 | <0.0001 | 0.0042 | <0.0001 | <0.0001 | <0.0001 | <0.0001 | <0.0001 |

* Selected countries are classified into different categories based on their death occurrence probability, $\pi(.)$ on 21st March, 2019

## 4. Discussion

From an extensive reporting of data on COVID-19 from different regions of the world, 75% of COVID-19 infection cases and 92% of deaths related to COVID-19 infections were registered in seven countries (Table 1). The situation of Italy was seems to be very critical due to increasing numbers of COVID-19 infected cases and its associated deaths in 51 days. But, in terms of mortality rates with respect to duration, Iran reached to 7.3% only in 31 days, whereas Italy took 51 days to reach to 8.6%, which is of serious concern. The chronicity and infectious nature of COVID-19 can be seen from the fact that within a span of seven days (50–57) the mortality rates varies from 2–9% as 2% (Germany), 3.6% (France), 4.4% (United Kingdom), 5% (Spain) and 8.6% (Italy). In Germany, where 0.2% mortality rate has reported in 54 days and belongs to Category I, the forecast presented in Table 2 and depicted in Fig 1, suggest that within a duration of 76 days with 95% CI (65–151) will raise to 1%, thereafter will require on

**Table 3. Duration of exposure and mortality rates in different countries due to COVID-19 on 28th March, 2020.**

| Country | Total Cases | Total Deaths | Duration of exposure (28th March, 2020) | Mortality rate $\pi$ | Category# |
|---|---|---|---|---|---|
| Germany | 48582 | 325 | 61 | 0.007 | I |
| China | 82213 | 3301 | 89 | 0.040 | II |
| **World** | **591971** | **27090** | **89** | **0.046** | **II** |
| France | 32964 | 1995 | 64 | 0.061 | III |
| Iran | 32332 | 2378 | 38 | 0.074 | III |
| Italy | 86498 | 9136 | 58 | 0.106 | III |
| Spain | 64059 | 4858 | 57 | 0.076 | III |
| United Kingdom | 14543 | 759 | 58 | 0.052 | III |

# Category I if $\pi \leq 2$%; Category II if $\pi$ (2–5)%; Category III if $\pi > 5$%;

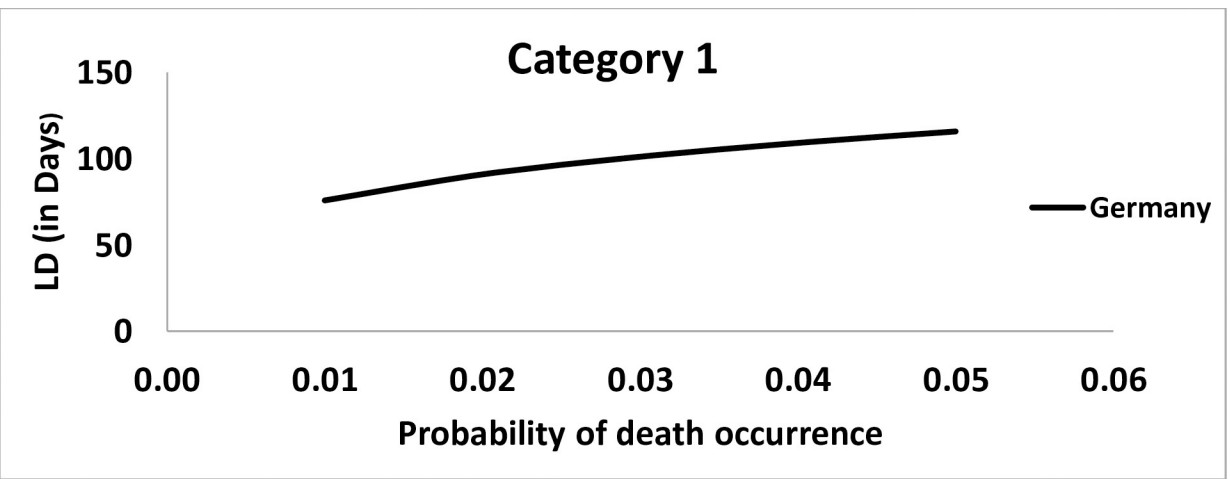

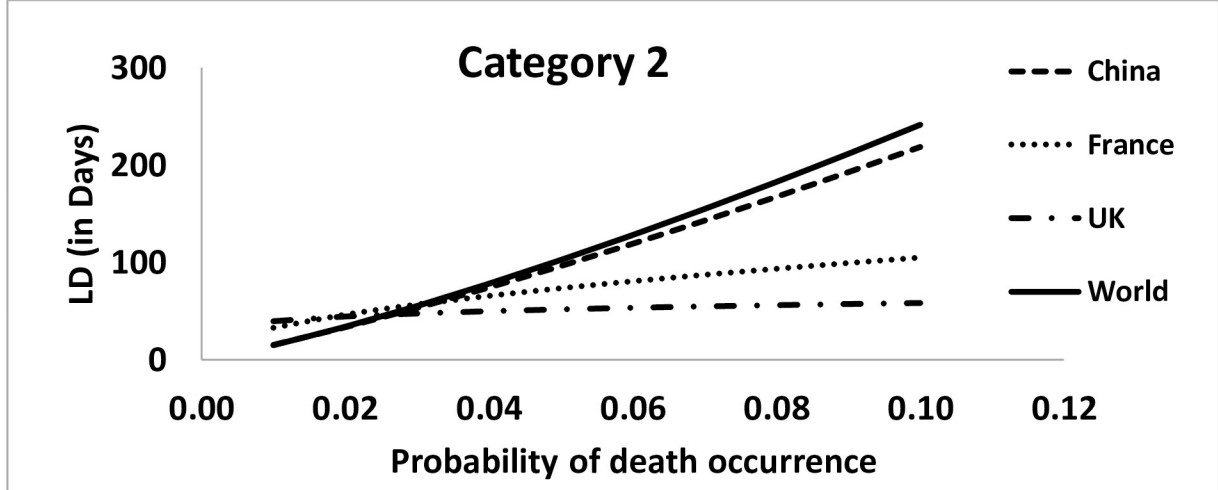

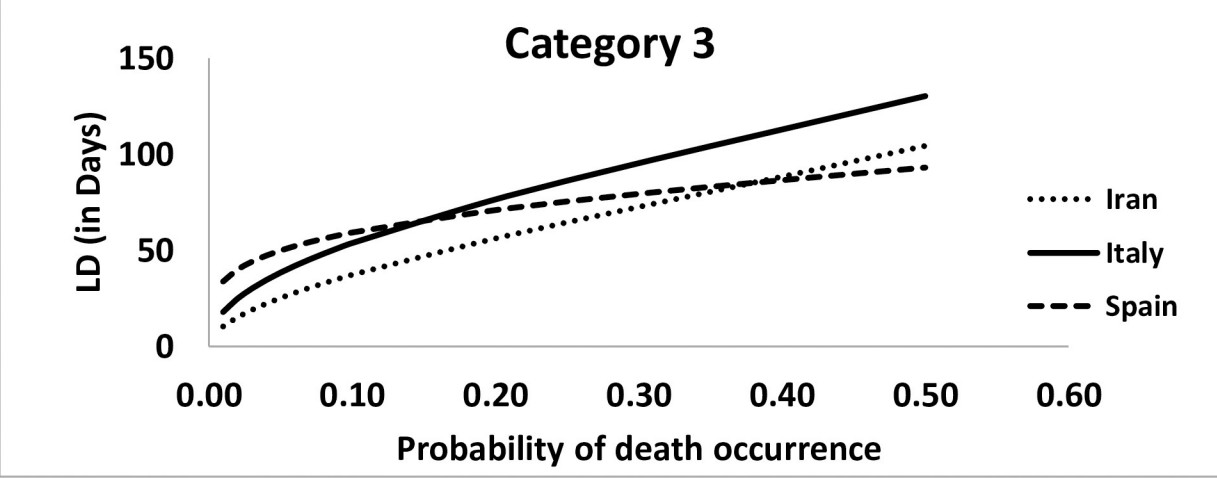

**Fig 1. Pattern of lethal duration of exposure to COVID-19due to probability of deaths in different categories of countries.**

an average an additional 15 days to reach to 2% deaths and if same trend follows, an additional 25 days will raise the mortality rate to 5%. Here, lower limit of the 95% confidence interval, for

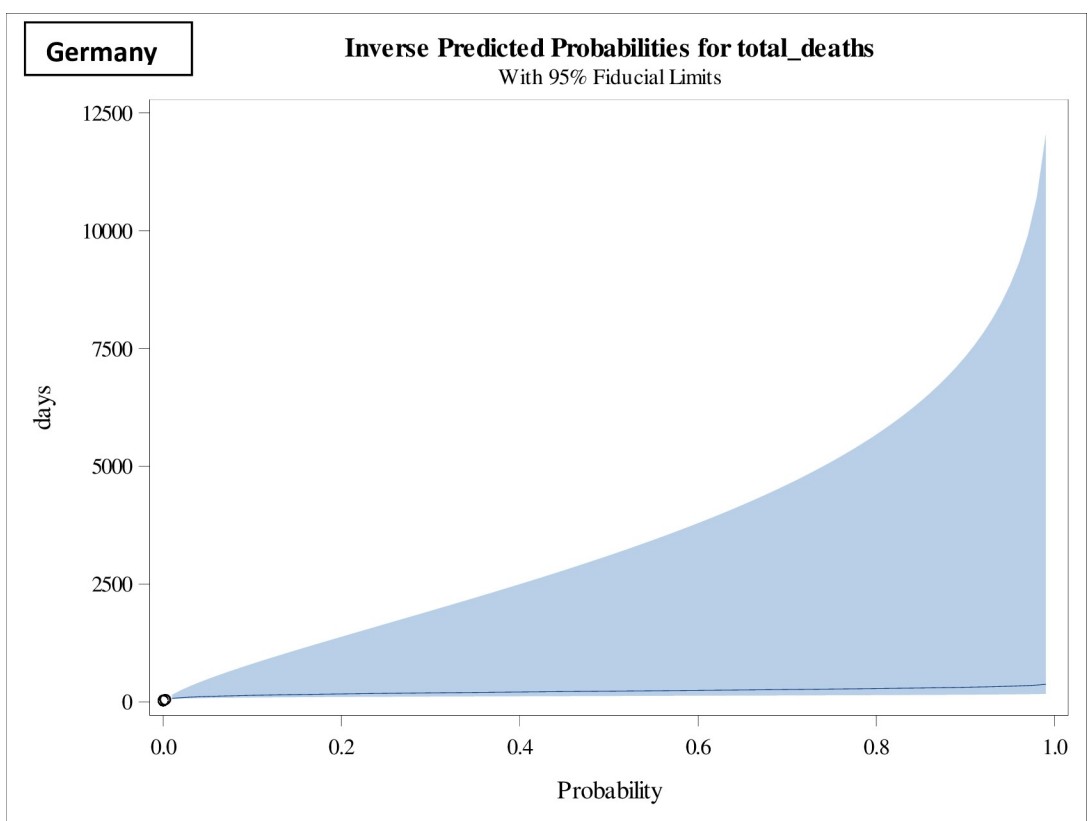

**Fig 2. Inverse predicted probabilities of total deaths in Germany of Category I having Class 1 (0.01–0.05) mortality rates.**

example, in case of Germany was 65 days, is indicating that if situation persists and remains uncontrolled even after the lockdown or social distancing, the mortality rate will increase to 1% before the scheduled pattern and the upper limit, says, for example 151 days, shows the effectiveness of initiatives towards controlling the situation. This duration found is statistically associated with predicted mortality rate (p-value: 0.0021), which means that an increase in lethal duration increases the predicted mortality rate [12, 13]. To validate the prediction and performance of the model, we used the 28th March, 2020, with reference to Germany, as predicted, within 7 days, i.e., in 61 days of exposure, the mortality rate increased from 0.2% to 0.7%.

In case of Categories II countries, viz., France and United Kingdom, where deaths reported as 3.6% and 4.4%, respectively, the forecast in Table 2 and Fig 2, suggest that the rates will increase to 6% in lethal duration of exposure of 80(67–216) days in France and 53(52–55) days in United Kingdom. The durations was significantly associated with predicted probabilities of deaths and found to be statistically significant for France and United Kingdom; p-value: 0.0042 and <0.0001, respectively. Obtained intervals (Table 2) have found to be consistent with the death rates observed in Table 3 on 28th March, 2020for France as 6% on 64th day and United Kingdom as 5.4% on 58th day.

Category III countries, viz., Iran, Italy and Spain, mortality rates on 21st March, 2020 were, 7.3%, 8.6% and 5%, respectively. The projected lethal duration of exposure for raising or mortality rates to 10%, as presented in Table 2 and Fig 3, in these counties shown that Iran will take on an average of 37(36–39) days; Italy will take 54(52–56) days and Spain will take 59(58–61) days, which is quite close to the mortality rates of Table 3 observed mortality

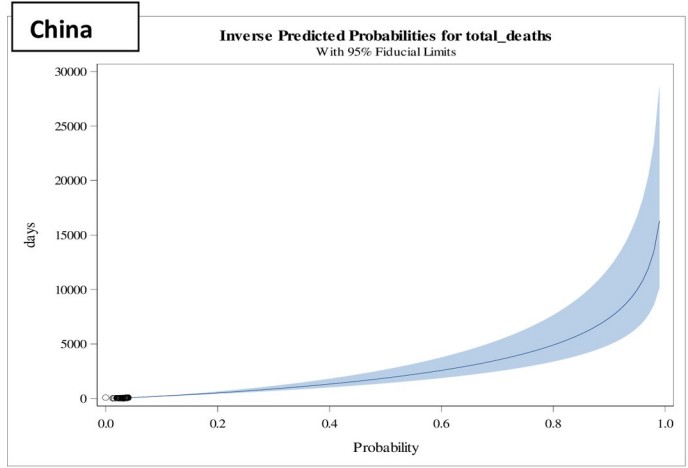

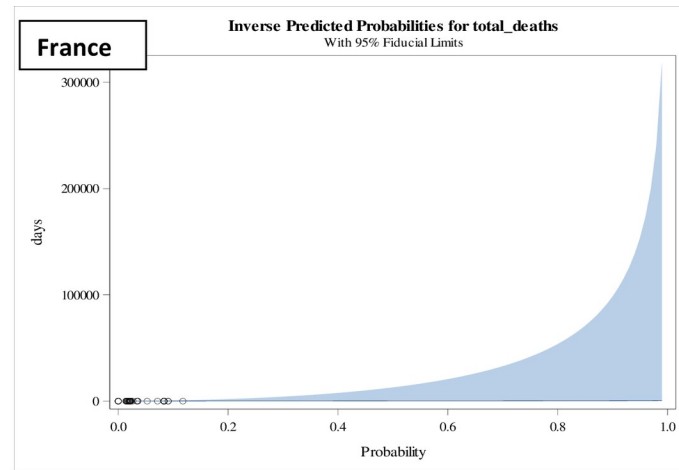

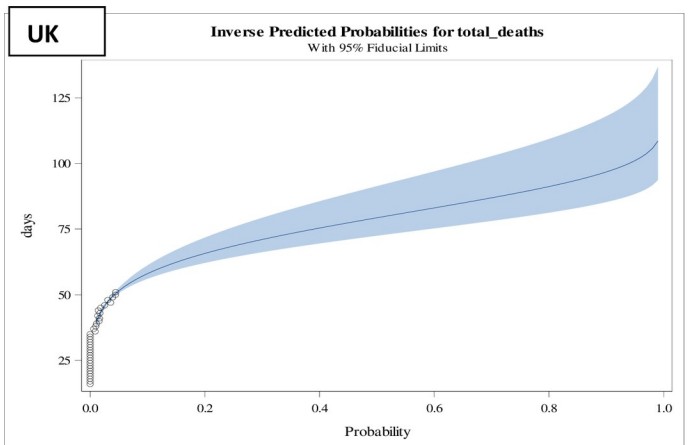

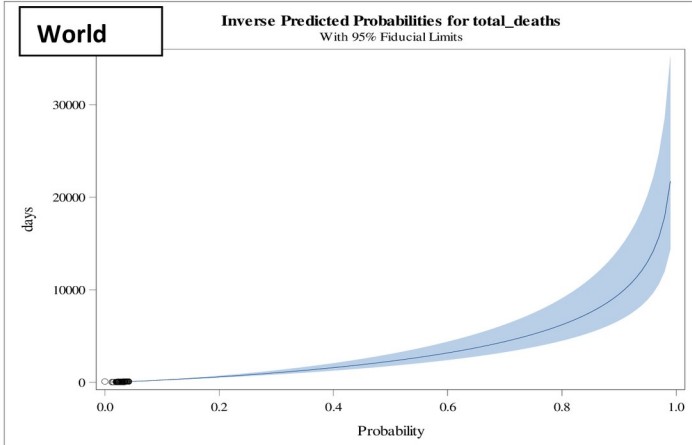

**Fig 3. Inverse predicted probabilities of total deaths in counties, China, France, United Kingdom (UK) and World as a whole of Category II having Class 2 (0.06–0.1) mortality rates.**

rates on 28th March, 2020, as Iran 7.4% on 38 days, Italy 10.6% on 58th day, and Spain 7.6% on 57th day. The duration was significantly associated with predicted mortality rates in these countries and found to have similar trend as seems in countries from category I and category II (Table 2).

## 5. Conclusion

The projection of lethal duration of exposure to COVID-19 for different countries, where comparatively higher number of infected cases and deaths were reported, is found to be very realistic and effective to predict the mortality rates and suggest higher chances of increase in rates. The present study is adding information about the quantum of the severity of COVID-19 and time elapsed to death that can occur based on the current trend, which will help in planning strategies required to be implemented in time. The obtained estimates of lethal duration of exposure give an indication of the casualty rates across the spectrum of COVID-19 disease and would eventually help to implement appropriate strategies to tackle this pandemic by taking necessary and appropriate steps such as precautions, self-sanitization and isolation, and by practicing social distancing.

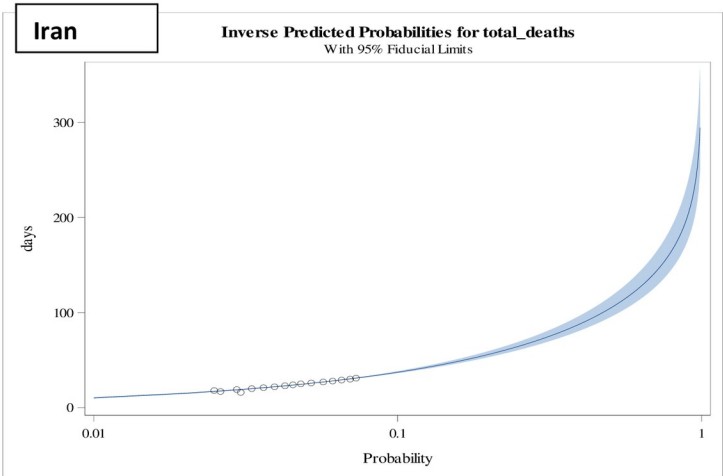

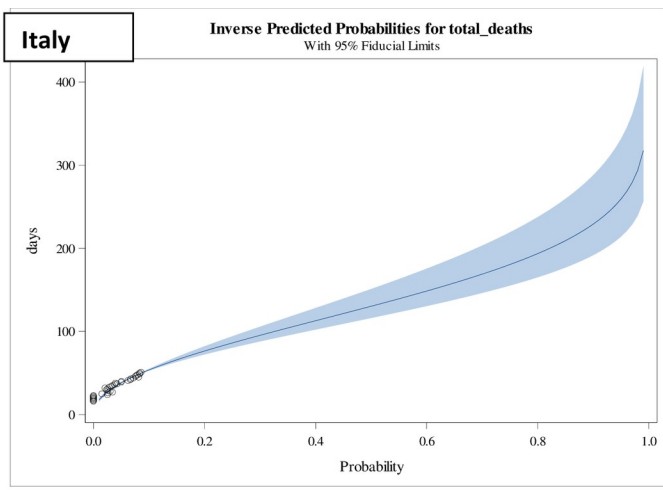

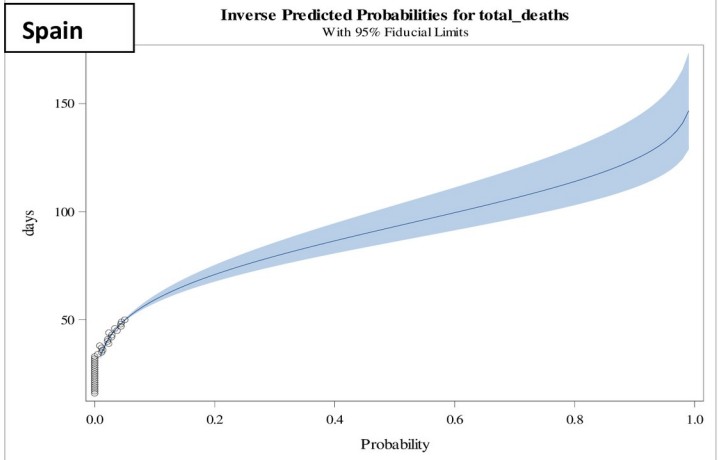

**Fig 4. Inverse predicted probabilities of total deaths in counties, Iran, Italy and Spain of Category III having Class 3 (0.2–0.5) mortality rates.**

## Supporting information

**S1 Appendix.**
(DOCX)

## Acknowledgments

The authors are very grateful to Prof Russell Kabir, Prof. Baki Billah, Prof. Kannan Navaneetham, the anonymous referee and the editorial board for their constructive and valuable comments, which helped to improve the contents of the paper.

## Author Contributions

**Conceptualization:** Vivek Verma.

**Data curation:** Anita Verma.

**Formal analysis:** Vivek Verma.

**Investigation:** Anita Verma, Dilip C. Nath.

**Methodology:** Vivek Verma.

**Project administration:** Ramesh K. Vishwakarma.

**Resources:** Dilip C. Nath.

**Software:** Ramesh K. Vishwakarma.

**Supervision:** Dilip C. Nath.

**Validation:** Hafiz T. A. Khan.

**Visualization:** Vivek Verma.

**Writing – original draft:** Vivek Verma, Ramesh K. Vishwakarma, Anita Verma, Hafiz T. A. Khan.

**Writing – review & editing:** Dilip C. Nath, Hafiz T. A. Khan.

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
