## [Decision Letter · Decision Letter 0]

24 Apr 2020

PONE-D-20-10057

Time-to-Death approach in revealing Chronicity and Severity of COVID-19 across the World

PLOS ONE

Dear Dr Verma,

Thank you for submitting your manuscript to PLOS ONE. After careful consideration, we feel that it has merit but does not fully meet PLOS ONE’s publication criteria as it currently stands. Therefore, we invite you to submit a revised version of the manuscript that addresses the points raised during the review process.

This paper is very relevance in the context of ongoing pandemic. The reviewers have suggested some corrections. Their comments are appended below. In particular, the objective section is too long. Some of them are rational and can be merged with the introduction. Objective section should be brief and clear. Similarly, the method section can be replaced with Data and Method, which should have two sub-section of Data and Method. As reviewer suggested, clearly specify the assumptions of the method.

We would appreciate receiving your revised manuscript by Jun 08 2020 11:59PM. To enhance the reproducibility of your results, we recommend that if applicable you deposit your laboratory protocols in protocols.io, where a protocol can be assigned its own identifier (DOI) such that it can be cited independently in the future. For instructions see: http://journals.plos.org/plosone/s/submission-guidelines#loc-laboratory-protocols

We look forward to receiving your revised manuscript.

Kind regards,

Kannan Navaneetham

Academic Editor

PLOS ONE

Reviewers' comments:

Reviewer's Responses to Questions

**Comments to the Author**

1. Is the manuscript technically sound, and do the data support the conclusions?

Reviewer #1: Yes

Reviewer #2: Yes

Reviewer #3: Yes

2. Has the statistical analysis been performed appropriately and rigorously? 

Reviewer #1: Yes

Reviewer #2: Yes

Reviewer #3: I Don't Know

3. Have the authors made all data underlying the findings in their manuscript fully available?

Reviewer #1: Yes

Reviewer #2: Yes

Reviewer #3: Yes

4. Is the manuscript presented in an intelligible fashion and written in standard English?

Reviewer #1: Yes

Reviewer #2: Yes

Reviewer #3: Yes

5. Review Comments to the Author

Reviewer #1: The research topic is very relevant contemporary. The findings of the research will make a good contribution to understand the situation of current pandemic. The title of the research is concise and indicates the content of the research. The abstract is well structured and presents the relevant information. The background provides a good contextual information about the research with a proper logical flow and clear expression and very good use of professional language. Method section is very well described and all the stages of data analysis are clearly explained. Statistical analyses are performed accurately. The findings of the research will have a huge impact for future research development. Ethical issues are considered and concluding remarks are inline with the research findings.

Reviewer #2: A very well written paper that proposed a modelling to predict the duration of exposure to COVID-19 and its impact on mortality in several countries and then across the globe. This idea can be adopted by other affected countries in the world and also for any other future outbreak. This sort of finding is very useful to health service authorities in planning urgent strategies to be implemented on a timely manner to tackle the burden of the outbreak. The model was developed and then validated on a hold out data, which shows that model's prediction performance is very good.

However, I have a few queries to authors:

(a) what are the underlying assumption/s of this modelling technique; please address them (if any) clearly in the method section of the paper;

(b) across the globe the number of COVID-19 cases are dropping due to social distancing and other measures. My concern is whether the model can be useful to capture this downfall of confirmed cases.

Reviewer #3: Reviewer’s comments

Objective: It should be specific.

• The author mixed justification of the study and possible recommendation of the study under objective, e.g. first 12 sentences under the headline (Objective) belong to justification. And part of last sentence e.g. “Which will help in planning strategies required to be implemented in time”. It appears as a recommendation based on study findings.

Methodology:

• Based on mortality rate the author sets the affected countries into three category. As per categories, the predicted mortality rates were grouped into 3 groups. Group 1: 1 to 5%, Group 2: 6 to 10% and group 3: 20 to 50%. It is not clear to me what about 11 to 19% is.

• Methodology means what the author does to conduct the study. Therefore, first two sentences in page 6 may be reformulated.

Result:

• 7th line Page 6, it will be 28th.

• Group 3: 20 to 50%. Percentage of 11 to 19 is missing.

• 11th line “For category I country, the prediction was ……………………………………………………………. having mortality rates of 20-50%.” It is same as methodology. Sentences needs to be modified if it is result.

I am not a statistician. Therefore, a statistician must critically analysis statistical part e.g.:

• Statistical formula

• Figure on inverse probability analysis for total deaths

• Pattern of probability death occurrence

Discussion:

• Discussion part may be shorten by omitting duplication of sentences used in result section.

• There is no reference in discussion. If similar study on prediction of lethal duration is conducted on other communicable diseases that may be used as reference for discussion.

Conclusions:

• Contents of conclusions should include only finding of the study. In one sentence future term is used that needs to be corrected.

• First sentence and third sentences of conclusions may be deleted.

• Part of last sentences e.g. “Would eventually help to implement strategies to tackle this pandemic by taking necessary appropriate steps.” Better to mention what are the necessary steps; if the author identified some of the necessary steps.

6. PLOS authors have the option to publish the peer review history of their article (what does this mean?). If published, this will include your full peer review and any attached files.

Reviewer #1: Yes: Russell Kabir

Reviewer #2: Yes: Baki Billah

Reviewer #3: No

---

## [Author Response · Author response to Decision Letter 0]

25 Apr 2020

Reviewer 2 Comments:

(a) what are the underlying assumption/s of this modelling technique; please address them (if any) clearly in the method section of the paper;

Response: The suggestion has discussed in 6th line of para 2 of page 6 of Section 2.

(b) across the globe the number of COVID-19 cases are dropping due to social distancing and other measures. My concern is whether the model can be useful to capture this downfall of confirmed cases.

Response: The aspect of social distancing and other measures are taken care while defining the 95% confidence interval for the lethal duration. The suggestion has discussed in 13th line of para 1 of page 7 of Section 4.

Reviewer 3 Comments:

1. Objective: It should be specific.

• The author mixed justification of the study and possible recommendation of the study under objective, e.g. first 12 sentences under the headline (Objective) belong to justification.

Response: The suggestion has incorporated in third and first and second paras of page 5 in Section 1 of the revised manuscript.

2. • And part of last sentence e.g. “Which will help in planning strategies required to be implemented in time”. It appears as a recommendation based on study findings.

Response: The suggestion has incorporated in the revised manuscript at 5th line of fourth para in page 8 of Section 5.

3. Methodology:

• Based on mortality rate the author sets the affected countries into three category. As per categories, the predicted mortality rates were grouped into 3 groups. Group 1: 1 to 5%, Group 2: 6 to 10% and group 3: 20 to 50%. It is not clear to me what about 11 to 19% is.

Response: To address the change in mortality rates, the classes were defined with either 1% or 10% increment in mortality rates, which has been incorporated in first para of page 6 in Section 2 of the revised manuscript.

4. • Methodology means what the author does to conduct the study. Therefore, first two sentences in page 6 may be reformulated.

Response: An appropriate changehas been incorporated in second para of page 6 in Section 2 of the revised manuscript.

5. Result:

• 7th line Page 6, it will be 28th.

Response: Done in the revised manuscript.

6. • Group 3: 20 to 50%. Percentage of 11 to 19 is missing.

Response: To address the change in mortality rates, the classes were defined with either 1% or 10% increment in mortality rates, which has been incorporated in first para of page 6 in Section 2 of the revised manuscript.

7.• 11th line “For category I country, the prediction was…………………. having mortality rates of 20-50%.” It is same as methodology. Sentences needs to be modified if it is result

Response: Done in the revised manuscript.

8. I am not a statistician. Therefore, a statistician must critically analysis statistical part e.g.:

•Statistical formula

•Figure on inverse probability analysis for total deaths

• Pattern of probability death occurrence

Response: Both Reviewers 1 and 2 found the statistical analysis has been performed appropriately and rigorously. 

9.Discussion:

• Discussion part may be shorten by omitting duplication of sentences used in result section.

Response: An appropriate change has been incorporated to the first para of page 7-8 at Section 4 of Discussion in the revised manuscript.

10. • There is no reference in discussion. If similar study on prediction of lethal duration is conducted on other communicable diseases that may be used as reference for discussion.

Response: Reference no [12-13] has been updated at first para of page 7 at Section 4 in the revised manuscript, which is related with forecasting and application of the method to antimicrobial agent’s activities. But no such work is found that is related with the prediction of lethal duration in connection with any communicable diseases.

11. Conclusions:

• Contents of conclusions should include only finding of the study. In one sentence future term is used that needs to be corrected.

Response: Done in the revised manuscript.

12. • First sentence and third sentences of conclusions may be deleted.

Response: Done in the revised manuscript.

13. • Part of last sentences e.g. “Would eventually help to implement strategies to tackle this pandemic by taking necessary appropriate steps.” Better to mention what are the necessary steps; if the author identified some of the necessary steps.

Response: Section 5 of Conclusion has been revised.

Note: All of the changes in the revised manuscript are highlighted using YELLOW colour.

---

## [Editor Report · Decision Letter 1]

29 Apr 2020

Time-to-Death approach in revealing Chronicity and Severity of COVID-19 across the World

PONE-D-20-10057R1

Dear Dr. Verma,

We are pleased to inform you that your manuscript has been judged scientifically suitable for publication and will be formally accepted for publication once it complies with all outstanding technical requirements.

With kind regards,

Kannan Navaneetham

Academic Editor

PLOS ONE
---

## [Editor Report · Acceptance letter]

4 May 2020

PONE-D-20-10057R1 

Time-to-Death approach in revealing Chronicity and Severity of COVID-19 across the World 

Dear Dr. Verma:

I am pleased to inform you that your manuscript has been deemed suitable for publication in PLOS ONE. Congratulations! Your manuscript is now with our production department. 

With kind regards,

on behalf of

Professor Kannan Navaneetham 

Academic Editor

PLOS ONE